# Measuring Quality of Life in Ovarian Cancer Clinical Trials—Can We Improve Objectivity and Cross Trial Comparisons?

**DOI:** 10.3390/cancers12113296

**Published:** 2020-11-07

**Authors:** Gita Bhat, Katherine Karakasis, Amit M. Oza

**Affiliations:** Princess Margaret Cancer Centre, University Health Network, Toronto, ON M5G 1X6, Canada; Gita.Bhat@uhn.ca (G.B.); katherine.karakasis@uhn.ca (K.K.)

**Keywords:** cancer, ovary, quality of life, patient reported outcomes, endpoints, treatment benefit, frequency of reporting

## Abstract

**Simple Summary:**

Regarded as the “disease that whispers” ovarian cancer remains difficult to diagnose and treat due to other complex conditions that may occur together. As research into new treatments for ovarian cancer continues, growing efforts to understand how treatments may be impact the day to day lives of women is also growing. Known as Patient Reported Outcomes or PROs and quality of life (QoL), the research community is improving how this important information communicated by women with ovarian cancer is captured across their cancer journey, and importantly, how this information can be used further refine treatments. Understanding the impact treatment has on day to day living is important, as new treatments should not only help control cancer cells, but also help keep women living with ovarian cancer living their lives to the fullest.

**Abstract:**

Epithelial ovarian cancer (EOC) remains a lethal disease for the majority of women diagnosed with it worldwide. For the majority of patients, diagnosis occurs late, in the advanced setting. Disease-induced as well as treatment-related adverse events can negatively impact quality of life (QoL). Research to date has captured these data through use of patient-related outcomes (PROs) and, increasingly, has become an area of increased attention and focus in clinical trial reporting. QoL/PRO measurements in EOC clinical trials at different transition points in a patient’s journey are increasingly being recognized by patients, clinicians and regulatory agencies as the key determinants of treatment benefit. Various context-specific PROs and PRO endpoints have been described for clinical trials in EOC. Standardized approaches and checklists for incorporating PRO endpoints in clinical trials have been proposed. In a real-world clinical practice setting, PRO/QoL measures, which are meaningful, valid, reliable, feasible and acceptable to patients and clinicians, need to be implemented and used. These may assist by serving as screening tools; helping with the identification of patient preferences to aid in decision making; improving patient–provider communication; facilitating shared decision making. Importantly, they may also improve quality of care through an increasingly patient-centered approach. Potential areas of future research include assessment of anxiety, depression and other mental health issues. In good prognostic groups, such as maintenance clinical trials, following patients beyond progression will capture possible downstream effects related to delaying the psychological trauma of relapse, symptoms due to disease progression and side-effects of subsequent chemotherapy. Identifying PRO endpoints in next-generation-targeted therapies (including immunotherapies) also warrants investigation.

## 1. Challenges of Managing Ovarian Cancer

Epithelial ovarian cancer remains a significant cause of morbidity and mortality worldwide with more than 295,000 women diagnosed and in excess of 180,000 deaths annually (GLOBOCAN 2018). More than half of the women have advanced disease at the time of diagnosis, and almost three-quarters will unfortunately succumb to it within five years of diagnosis [1]. The cornerstone of systemic treatment for advanced disease in the front-line setting is platinum-based chemotherapy in combination with a taxane; unfortunately, the majority of patients will experience recurrence [1]. Recurrent ovarian cancer is incurable, and disease and treatment cause a plethora of symptoms that impact on quality of life (QoL). Clinical trials incorporate validated objective measures of quality of life, but inter-trial comparisons are challenging as trials variably incorporate, analyze and present these findings [1,2,3,4].

There are multiple factors that impact reporting and interpretation of QoL including cancer type, treatment- and patient-specific factors [5]. Furthermore, QoL may also be impacted by age at diagnosis, especially loss of child-bearing potential for younger women and the time-point in the disease trajectory [5]. While a higher level of toxicity is acceptable in first-line chemotherapy where cure or prolonged disease control is being sought, after making allowances for performance status and age, the focus shifts to improving symptom control in the setting of recurrent disease. Therefore, the effect of therapeutic interventions on QoL has to be borne in mind, especially in patients with recurrent disease, for whom the intent of treatment is palliative [2].

This review focuses on estimating QoL in ovarian cancer, with a specific lens on the historical approach, implications of incorporation within contemporary clinical settings and future opportunities to expand inclusion in clinical trials and practice.

QoL, health-related quality of life (HRQoL), patient-reported outcome measures (PROMs) and patient-centered outcomes (PCOs) are unique terminologies. The World Health Organization (WHO) has characterized QoL as reports from the perspective of an individual’s perception [6,7]; whilst a nuanced iteration of this through the health-related quality of life (HRQoL) tools is the capture of duration of life as impacted and influenced by an individual’s disease [8]. These tools attempt to capture how individual health may be ultimately influenced by the care received [8]. The third most commonly utilized assessment tool is the patient-reported outcome measures (PROMs), which captures status reported by the patient alone, without interpretation of the response by a health care provider [9,10]. Therefore, whilst all are contemporary tools to record and objectively facilitate data capture, they are not interchangeable [9,10].

Patient-reported outcomes, such as symptoms, functional status, treatment adherence and satisfaction with care (efficacy and safety), help clinicians to better define the drug profile [7]. Data are captured by way of questionnaires completed by the patient [11] or via interviews [11], which are validated and standardized methodologies for this area of research [11]. PROs have evolved to support specific disease settings such as generic PROMs designed with the objective of capturing data across disease groups [11], and conversely, disease-specific PROMs supporting focused inquiry of a group of patients with a specific disease [11]. Considered an umbrella term, HRQoL encompasses a variety of outcomes reported by the patient including symptom burden and psychological well-being [11]. PRO is a generic term that refers to the patient’s perspective of self. The FDA now actively encourages the use of PROs in clinical trials [12]. Patient-centered outcomes (PCOs) are defined as outcomes that are important to patients and caregivers [13]. PCOs include patient satisfaction (e.g., the QLQ-INPATSAT32 (Inpatient Satisfaction with care) questionnaire to measure patient satisfaction with inpatient care [14]), decision regret (e.g., five-item decision regret scale used in surgery and adjuvant therapy for breast and prostate cancers [15,16]), patient preference (e.g., quality adjusted life years [17]) and HRQoL (e.g., the European Organization for the Research and Treatment of Cancer Quality of Life Questionnaire (EORTC QLQ-C30) questionnaire, which has been described in detail in this review) [17].

## 2. Why Measure Quality of Life?

As patient-centredness is increasingly identified as a critical component of quality health care; measurements of QoL and PROMs in ovarian cancer clinical trials at different transition points in a patient’s journey are being recognized by patients, clinicians, regulatory agencies and other health care stakeholders as key determining factors of treatment benefit. QoL endpoints have been identified as an endpoint of interest in the recurrent disease setting during the Fifth Ovarian Cancer Consensus Conference [3] which will be discussed in detail in this article [3]. Complementary findings by Donovan et al. show that the key domains of interest in patients with ovarian cancer are specific abdominal symptoms, weight issues and sexual functioning [18].

Baseline or pre-treatment QoL has been shown to be a prognostic as well as a predictive indicator of ovarian cancer [19,20,21]. In advanced ovarian cancer, the European Organization for the Research and Treatment of Cancer Quality of Life Questionnaire (EORTC QlQ-C30) is predictive of overall and progression free survivals (OS/PFS) in advanced ovarian cancer when examining baseline data [19]. When compared with performance status and grade, global QoL reported with this tool were predictors of OS even three months after completion of chemotherapy [19]. Complementary findings have been reported when examining baseline data in the health, physical and well-being domains [19,20,21]. Improvements in appetite, constipation and global health scores during the first three months of treatment of ovarian cancer are also associated with a significant improvement in survival time [22].

Despite the availability of data showing the importance of QoL in ovarian cancer and the availability of guidelines on incorporating PROs in clinical trials, there has been variability in the design, reporting and interpretation of PRO endpoints in clinical trials [23]. Only a third of 50 randomized clinical trials in patients with gynecologic cancers provided reliable data to fully appreciate the value of PROs in decision making [24]. Only 16% of the studies reported PROs as the primary endpoint; the methods of collecting PRO data and the modes of administration of instruments were only described in 16%; PRO hypothesis was mentioned in 14% and statistical approaches for dealing with missing data were described in 18% [2]. Less than one third (30%) discussed the clinical significance of HRQoL findings [24]. In a systematic review of HRQoL reporting in ovarian cancer Phase III trials, which included 35 studies, it was shown that the inclusion of QoL assessments from 1980 to 2010 and beyond increased from 2% to 62%, respectively [2].

## 3. Steps Involved in Integrating PROs in Ovarian Cancer Clinical Trials

It is important to refine the inclusion and reporting of QoL activities. A multistep process, initial design and development of QoL inquiry includes careful analysis of context and hypothesis to ensure appropriate analytical tools and statistical analyses are selected [3]. This will help ensure setting the context occurs to accurately reflect the patient population, defining treatment aims and objectives, as well as laying emphasis on patient preferences and measuring factors of importance to patients. As such, due consideration must be given to whether results will impact regulatory approval and/or clinical practice [3]. These activities help frame the PRO hypothesis—prioritizing and selecting PROs to appropriately support primary trial endpoints [3]. Methodology helps bring cohesive inquiry across several aspects of QoL assessments such as determining which is/are the right instrument(s) to be used. Typically, questionnaires with multiple domains are required for early phase drug development, when focus is on the signal of activity rather than efficacy; however, if focus is on domains more affected by disease, a multi-domain questionnaire is appropriate. When needs and expectations of patients are better known in later trial phases, use of fewer domains may help interpret changes better [7].

### 3.1. Measuring General Health Status in the Context of Ovarian Cancer: EQ-5D-5L and EQ-VAS

The EQ-5D-5L validated HRQoL tool has five domains, each with five possible responses: no problem (level 1), slight (level 2), moderate (level 3), severe (level 4) and extreme problems (level 5), illustrating a range of patient-reported measures [25,26]. For example, topics of self-care, mobility and anxiety or depression can be reported upon, with the resulting five levels combined to create a five-digit number illustrating a patient’s particular health state (e.g., 31234) [26]. This is then transformed to a health utility index (HUI) value, determined from the health states using the standard value sets produced with the EuroQol Group’s standardized valuation technology [26]. EQ-VAS is a self-reported vertical visual analogue scale that measures current health status on a scale from zero (worst imaginable health scale) to 100 (best imaginable health scale) [27]. It is included in the EQ-5D-5L and provides important, complementary information on patients’ views about their own health [4,27].

### 3.2. Cancer- and Disease-Specific Questionnaires: EORTC and FACIT

Questionnaires produced by EORTC and the Functional Assessment of Chronic Illness Therapy (FACIT) are widely used in oncology. The core measures, the EORTC QLQ-C30 (30 items) and Functional Assessment of Cancer Therapy–General (FACT-G) (27 items), include questions on overall quality of life and health and cover the domains of physical, emotional, role/functional and social functional/well-being. Designed with the objective of capturing a patients’ capacity to fulfill the activities of daily living, the EORTC QLQ-C30 is a 30-item cancer-specific questionnaire that converges functioning (five elements), symptom (eight elements) and a global scale predicated on two main questions being addressed by patients—(i) How would you rate your overall health during the past week?; (ii) How would you rate your overall quality of life during the past week [28]? Tested for reliability and validity [28,29,30,31], the EORTC QLQ-C30, when transformed, illustrates higher scores in global and functioning scales, indicating better QoL, with magnitudes of change greater than 10 points being suggestive of changes in patient perspective that are clinically significant [29]. The QLQ-C30 also includes subscales on cognitive functioning (two items) and financial impact (one item). It places greater emphasis on symptom management with subscales on pain, fatigue, nausea/vomiting (two, three and two items, respectively) and dyspnea, insomnia, appetite loss, constipation, and diarrhea (one item each) [28]. On the QLQ-C30, subscales are scored and reported separately to a total score of 15. The scoring algorithm does not include a total score that is summed over all items and domains. Instead, a summary HRQoL score is derived from the two global quality of life and health questions. Hence, although the QLQ C-30 provides comprehensive coverage, it yields a large number of outcome variables resulting in problems with multiple comparisons [32]. In contrast, the FACIT scoring algorithm includes a total core score (FACT-G score = sum of all 27 core items) as well as subscales for the four well-being domains of physical, functional, emotional and social/family (PWB (Physical Well-Being), FWB (Functional Well-Being), EWB (Emotional Well-Being) and SWB (Social Well-Being) The FACIT (Functional Assessment of Chronic Illness Therapy) disease-, symptom- and treatment-specific instruments include these 27 core items plus several “additional concerns”, which are scored as a separate subscale and added to the FACT-G total score to give a total score (e.g., FACT-O score) [33]. FACIT has an optional Trial Outcome Index (TOI) designed to be maximally sensitive and responsive to treatment effects. The TOI is the sum of the PWB, FWB and relevant additional concerns subscale [28,32,33].

The FOSI (Functional Assessment of Cancer Therapy—Ovarian Symptom Index) is a validated tool with eight items that measure response to treatment based on symptom assessment [34]. It takes into account a subset of questions from the FACT-O questionnaire and has been validated in a population of 62 patients with advanced ovarian cancer [34]. The questions assess pain, fatigue, nausea, vomiting, bloating, cramping, worry, and QoL. Patients report their symptoms over the past week using a five-point Likert scale, which ranges from 0 (not at all) to 4 (very much) [34]. The FOSI score is calculated by multiplying the total score by 8 and dividing the result by the number of responses [34].

The Disease-Related Symptom score is a subset of the National Comprehensive Cancer Network Functional Assessment of Cancer Therapy Ovarian Symptom Index–18 (NFOSI-18), which evaluates nine symptoms related to disease or treatment [35]. Scores range from 0 to 36, with higher scores indicating a lower burden of symptoms. A three-point difference was defined as clinically meaningful [36].

Developed and validated by the Gynecologic Cancer InterGroup (GCIG), the Measure of Ovarian Symptoms and Treatment (MOST) PROM is designed for use during chemotherapy for recurrent disease, with the objective of quantifying symptom burden, adverse events and potential symptom improvement [37]. As such, it is primarily indicated for use in palliative chemotherapy. The MOST version 2 has 24 items and yields five indices: MOST-Abdo (abdominal symptoms), MOST-DorT (disease- or treatment-related symptoms, MOST-Chemo (assessment of the burden of chemotherapy toxicities), MOST-Psych (prospective assessment of change in anxiety and depression from baseline) and MOST-Well-being (prospective assessment of change in perceived well-being from baseline) [37]. While this is a flexible instrument that can be modified depending on the aims and objects of the trial, its target population, treatments and PRO hypothesis, test–retest reliability and discriminative validity and responsiveness of MOST-Psych are pending assessment [37]. It can be completed within five minutes and reviewed by physicians and compared with previous assessments within one minute [37].

### 3.3. Measuring QoL in Clinical Trials at Different Transition Points in a Patient’s Journey: One Size Does Not Fit All

PROMs should be context specific with respect to clinical scenario, patient population, treatment intent and pre-defined endpoints [3]. Careful consideration must be given to the PRO hypothesis and how PRO endpoints can underpin and support the primary/co-primary endpoint. Utility measures such as quality-adjusted time without symptoms or toxicity (QTWiST), quality-adjusted PFS (QAPFS) should be used wherever relevant [3]. With Q-TWiST, completion by the patient or the clinical team generates a score, which reflects quality of life-adjusted weighted sums representative of mean durations of health states [38]. For example, the duration of survival across clinical health states, including: treatment-related adverse events, disease progression or progression free periods; spanning perfect health (1.0) and death (0) [38]. Accompanying this data is QAPFS—associated with progression free survival states—calculated by overall health utility weights with state of mean PFS time [39]. The GCIG Ovarian Cancer Consensus Conference has recommended that the overall survival (OS) is the ideal primary endpoint for first-line trails in ovarian cancer. However, PFS, if used, it should be supported by endpoints such as time to first subsequent treatment or beyond, relevant PROs and severity of toxicities [3]. Ideally, in platinum sensitive ovarian cancer, PFS supported by PROs should be the primary endpoint [3]. PROs should be a primary/co-primary endpoint along with PFS in platinum resistant or refractory ovarian cancer [3].

### 3.4. Defining Criteria for What Constitutes a Minimal Clinical Important Difference (MCID)

MCID is a context-specific, patient-centered concept [40]. It underpins the idea that the amount of improvement that is important to the patient must be determined while assessing the clinical utility of treatments intended to improve subjective outcomes [40]. ecorded scores are representative of changes in outcome that are perceived as meaningful to the patient [29,41]. It has been suggested that for a normalized scale, 5–10%, 10–20% and >20% change constitute small, moderate and large differences, respectively [29].

### 3.5. Defining the Statistical Plan

This includes assessment of whether the trial is adequately powered for PRO/QoL endpoints and if there is a statistical analysis plan in place. It is important to select the PRO instrument of choice and statistical analysis with appropriate endpoints to complement and support study objectives and hypothesis [3].

### 3.6. Specific Monitoring Plan

This involves having a strategy to minimize missing data such as training dedicated personnel for data collection and enabling patients to complete the questionnaires in the most convenient way (online, on a tablet or on paper, either at home or during clinic visits) [3]. This also includes steps taken to deal with missing data in the analysis—to investigate whether it was related to health status and discuss the potential impact of missing data on findings [3].

## 4. What Are the Implications for Patients, Clinicians and Regulatory Agencies?

There should be concordance and validity between PRO endpoints and true patient benefit, and measurement of objective benefit. Research is considered patient-centered if the focus includes outcomes that matter to the patients [13]. Quality of life tools and instruments are only valid if they accurately reflect patients’ subjective and objective symptoms and concerns, and are reliable.

The Ovarian Cancer National Alliance (OCNA) conducted a survey of women with gynecologic cancers to determine patient-identified surrogate endpoints and measures to determine the impact of QoL and treatment related adverse events [42]. This was accomplished using an anonymous online survey posted to www.ovariancancer.org incorporating anticipated questions regarding demographics, tumor data, patient preference regarding side effects and therapy endpoints [42]. Findings show minimum PFS or OS prolongation should be 5 months or longer for novel drugs to be deemed meaningful, 77% and 85% of the time, respectively [42]. The same study showed that the majority of women (55%) were interested in an agent that would produce disease stability without change in OS. The survey also showed that women were willing to accept greater toxicity (including grades 3 and 4) in the setting of front-line therapy with curative intent [42]. Interestingly, 44% of patients surveyed said that they would prefer a potential new therapeutic option that could lead to improvement in OS of five to six months, even with a three-fold higher rate of neurotoxicity [42]. Response patterns were not altered by recurrence status. This study also assessed patients’ perspectives on importance of financial impact within the context of decision-making [42]. While the majority of the respondents preferred no to minimal impact, few agreed to significant negative financial consequences [42]. Brown et al. surveyed the end-of-life preferences of gynecologic cancer patients [43]. They found that 78% of patients expressed their resolve to fight the disease should they be informed that current treatments were failing [43]. Similar findings were reported in Donovan’s study which showed that 25% of newly diagnosed ovarian cancer patients would opt for aggressive salvage intervention (median survival <1 week) than switch to palliative care [44].

Contemporary studies have examined several interrelated factors in women in the front-line advanced setting participating in randomized trials, which has shed important new light on HRQoL, social/economic factors and the impact of these within a larger context of determinants of health [45]. Moss et al. have shown that individual-level factors such as age, race, gender, cancer stage and toxicities, and social determinants such as socioeconomic status, were not statistically significantly associated with HRQoL [45]. However, when compared to women with health insurance covered by private entities, those who were not insured had a statistically significant lower mean HRQoL in the physical well-being, functional well-being, ovarian cancer-specific scores and TOI subscales. HRQoL was a predictor of OS, as women who had HRQoL data at 84 weeks were healthier with a better prognosis than other patients [46,47].

Regulatory agencies such as the United States FDA, European Medicine Agency and Health Canada have recognized the importance of PRO measures in the evaluation of cancer treatments. The FDA recommends an in-depth assessment of disease- and treatment-related issues beyond HRQoL [48,49,50,51,52,53]. The European Society for Medical Oncology (ESMO) and American Society of Clinical Oncology (ASCO) have emphasized the importance of accurately defining the clinical benefit of newer drugs to patients and have proposed standardized approaches to evaluate the outcomes of clinical trials by using the Magnitude of Clinical Benefit scale (MCB by ESMO) [54] or the Net Health Benefit (NHB by ASCO), [55]. These include survival endpoints in addition to adverse events and HRQoL [53,54,55].

## 5. Implication in Daily Clinical Practice

Detailed assessment of symptoms experienced by patients is an important cornerstone in clinical decision-making, especially in the setting of advanced or recurrent cancer [56]. Improvements in symptoms often indicate a response to treatment, whereas worsening symptoms may point to progressive disease or treatment toxicity. Clinical decisions such as a treatment change (e.g., reduction in chemotherapy dose for fatigue or neuropathy); administration of supportive care (e.g., providing an antiemetic for nausea, management of malignant bowel obstruction); triaging for additional medical services (e.g., psychosocial care, referral to the dietitian); or additional evaluation of a new or worsening symptom (e.g., imaging a patient with abdominal pain postoperatively or ahead of a scheduled interim assessment) may be made on the basis of patient-reported data [56].

This information is usually collected during a clinic visit and there may be discrepancies if the patient has forgotten to tell, the physician has forgotten to ask or take note, or if there was a time constraint [57]. This may necessitate the patient to call or have an unscheduled hospital visit with worsening of symptoms. In order to streamline the approach for patients and to systematically capture data, there has been a growing interest in using PROMs in routine practice as they can be used as guides to monitor patient symptoms—general, ovarian-cancer specific and those related to toxicity of ongoing and previous treatment [57]. Their use for clarifying patients’ needs and monitoring delayed side-effects is also being studied [57]. This approach improves physician–patient communication, and may provide more valid and comprehensive knowledge of patient’s problems when used as a dialogue tool along with blood tests and radiology [58]. Given its short questions, uniform response format and layout—which were designed keeping the ease of completion and rapid interpretation in mind—the application of MOST in clinical practice has also been considered during its development [37]. The role of MOST in daily clinical practice would be towards identifying the nature, severity and time course of symptoms and concerns, and the ability to access this information in real-time could improve decision making [37]. Data show that integration of electronic PRO into routine care of recurrent metastatic cancer is associated with survival improvement and may be considered as part of high-quality cancer care [59]. The potential mechanisms showed early responsiveness to patient symptoms, which prevented adverse downstream consequences and greater tolerance to chemotherapy [60].

With regard to the regulatory agencies, PROMs can provide important data on patient benefit and context that may allow some assessment between different drugs of the same pharmacological class. Robust data demonstrating a positive impact on the patient’s health status and daily life might facilitate inclusion in the reimbursement list [7] or compassionate use as in the case of Poly-ADP Ribose Polymerase (PARP) inhibitors.

## 6. Real-World Evidence

The ultimate goal of new treatments and interventions is to improve population outcomes. Real world evidence, which refers to population-based observational research, can help determine whether the anticipated benefits are being safely realized [61,62]. Surveillance, Epidemiology, and End Results (SEER; seer.cancer.gov) data show that, on average, newly diagnosed ovarian cancer cases have been declining at a rate of 2.5% annually over the last decade when age-adjusted [63] and are similar for death rates, which have declined at an average of 2.2% during the same period [63]. In Canada, the age-standardized incidence rate for ovarian cancer has decreased by 0.8% per year since 1992 and, similarly, the age-standardized mortality rate has reduced by 0.7% per year since 1974 [62]. The reduction in mortality rate has been attributed in part to the widespread use of protective oral contraceptive pills and decreased use of postmenopausal hormone supplementation [64,65]. In the absence of screening programs, which can yield lead time bias, improvements in relative survival or cancer-specific survival are plausibly related to treatment effect [62].

Potential contemporary maintenance therapy options are bevacizumab and PARP inhibitors [66]. As shown by Garofalo et al., roughly half (51%) of women who would benefit from maintenance did not receive it [67] and translating this to real-world scenarios, approximately half (56%) of BRCA mutation carriers received maintenance therapies [67]. Amongst patients who received maintenance therapy following platinum-based chemotherapy in second-line and beyond, 46% received a PARP inhibitor based regimen [67]. The published abstract does not mention reasons for under-utilization of maintenance options. Additional real-world analysis on the recurrence of ovarian cancer in *BRCA*-wild-type patients without maintenance conducted on 5535 patients has shown that 56% of *BRCA*-wild-type patients who received platinum-based chemotherapy without maintenance treatment relapsed within six months and were deemed to be platinum resistant [68]. PARP inhibitors have to show lack of a deleterious effect on QoL, and significantly prolong PFS, to be considered to be good maintenance options [66,69,70]. QoL data from these trials would reinforce the need to include PARP inhibitors in reimbursement lists and ensure their availability through extended access/compassionate access programs.

## 7. Clinical Benefit of Controversial First-Line Therapies for Advanced Stage Ovarian Cancer: Based on ESMO-MCBS Scores

The ESMO-magnitude of clinical benefit scale (MCBS)—standardized to assess the therapeutic benefit of anti-neoplastic therapies—considers the study of primary and secondary endpoints; absolute gains in OS and PFS and the bottom portion of the 95% confidence interval of the corresponding hazard ratio; as well as QoL or toxicity [71,72]. The data are analyzed with respect to the duration of response or survival in the control arm. This results in assignment of a clinical benefit ranking for the new drug [72].

ESMO-MCBS has been used to evaluate the evidence for current standard of care of newly diagnosed advanced ovarian cancer, with an emphasis on controversial therapeutic options, such as intraperitoneal chemotherapy, dose-dense paclitaxel and bevacizumab [73]. Based on these scores, it has been shown that dose-dense paclitaxel and intraperitoneal chemotherapy cannot be recommended as standard of care treatment. Bevacizumab should be reserved only for the high-risk population [73].

## 8. What Do We Know from Available Data in Ovarian Cancer Trials?

Effect of chemotherapy-induced peripheral neuropathy (CIPN) on HRQoL: CIPN has been reported to be the principal toxicity interfering with self-care, mobility and HRQoL [74]. Ezendam et al. assessed the prevalence of chemotherapy-induced peripheral neuropathy and its impact on HRQoL among ovarian cancer survivors [75]. Using the population-based Eindhoven Cancer Registry data [74], findings from a Dutch study of women newly diagnosed with ovarian cancer between 2000 and 2010 found 51% of participants reported neuropathy post-chemotherapy, characterized especially by tingling in hands and feet and numbness in fingers and toes, and 8% of women experienced neuropathic symptoms even twelve years after completion of treatment [75]. This was associated with worse functioning, overall HRQoL, pain and insomnia [75]. The data provide a picture of CIPN that mirrors the real-world scenario [75].

Conversely, in first-line trials the duration of assessment has been, on average, only six months after completion of chemotherapy [2]. Longer follow-up is critical to better understand the true extent of neuropathy and its impact on HRQoL. Data from two studies have shown reporting of mild to severe symptoms reported two years after completion of front-line chemotherapy and upwards of one-quarter of respondents experience mild symptoms [2,76]. Thus, longer follow-up is critical to better understand the true extent of neuropathy and its impact on HRQoL.

## 9. QoL Data from Anti-Angiogenic Trials in Ovarian Cancer

This section summarizes the QoL data from landmark clinical trials involving anti-angiogenics in ovarian cancer, in the front-line and recurrent settings.

First-line trials of Bevacizumab: The GOG-218 (NCT00262847) study [46] randomized patients who had undergone debulking surgery for advanced ovarian cancer to receive either chemotherapy alone (along with placebo), or chemotherapy along with bevacizumab (bevacizumab initiation) or chemotherapy along with bevacizumab followed by maintenance with bevacizumab (bevacizumab throughout). There was a modest improvement in PFS of 3.8 months for patients on maintenance bevacizumab [46]. QoL was measured using the FACT-O TOI questionnaire. No significant differences were found in the mean FACT-O TOI scores between the control (chemotherapy alone) group and the bevacizumab-throughout group after completion of chemotherapy [46].

To better understand the true prognostic value of QoL scores on survival in ovarian cancer, Phippen et al. [77] sought to evaluate the relationship between baseline and serial QoL measurements and survival in GOG-218 participants [77]. Over an exhaustive comparison of quartiles to baseline data, authors demonstrated baseline FACT-O TOI scores were independently prognostic of PFS and OS in advanced ovarian cancer [77]. In a separate analysis, Monk et al. [47] showed that Bevacizumab compromised QoL, to a mild extent during chemotherapy, but did not have prolonged effect after chemotherapy completion [47]. The authors concluded that while the initial detrimental effect on QoL by bevacizumab was statistically significant, it was unclear that it was clinically significant as well. It is posited that this finding may be attributed to the fact that the study was powered to support analysis of the PFS endpoint and may have been overpowered for the QoL endpoint [47].

ICON7 (NCT00483782) was another landmark trial in first-line treatment of ovarian cancer where patients were randomized to standard chemotherapy with paclitaxel and carboplatin alone vs. chemotherapy and bevacizumab followed by maintenance bevacizumab [78]. Patients with a poor prognosis (greater than 1 cm residual disease, inoperable stage III and stage IV disease) had the greatest benefit in terms of PFS and OS [78,79]. Stark et al. [80] analyzed the HRQoL outcomes from ICON7 using the EORTC QLQ-C30 questionnaire. Overall, global QoL improved during chemotherapy—for both groups, Bevacizumab maintenance was associated with a small but clinically significant decrement in QoL compared with standard chemotherapy. The possible causes of decrease in QoL were toxicity of bevacizumab, such as rash, hormonal symptoms, bleeding, thromboembolism or gastro-intestinal perforation [80], as well as the financial effects and impairment of women’s return to their life roles and responsibilities [80]. Therefore, balancing PFS prolongation and QoL must be considered within the context of routine clinical care [80].

GOG-218 (NCT00262847) and ICON7 (NCT00483782) showed a detrimental effect of Bevacizumab on QoL, despite improvements in survival [47,77,80]. Neither study captured patient preference (multiple intravenous infusions and resultant disruption in carrying out life roles) and anxiety arising out of the knowledge of the potentially life-threatening toxicities of bevacizumab such as pulmonary embolism, gastro-intestinal perforation and fistula formation. This aspect has been described in the section on missed opportunities.

### 9.1. First-Line Trial of Pazopanib

The AGO-OVAR16 (NCT00866697) study analyzed maintenance Pazopanib versus placebo in women who have responded to front-line treatment of advanced ovarian cancer [81]. While the primary study endpoint of PFS was met, there was no impact to OS [81]. The secondary HRQoL endpoints were the mean change in scores in those receiving maintenance Pazopanib and the impact of toxicities on HRQoL, which were measured using EORTC-QLQ-C30, QLQ-OV28 and EQ-5D-3L [81]. Friedlander et al. [23] presented the post-hoc exploratory analysis wherein it was shown that Pazopanib was associated with superior QAPFS (386 days with Pazopanib versus 359 days with placebo (*p* = 0.052)); disease progression was associated with a significant decline in HRQoL (*p* < 0.0001) and Pazopanib maintenance resulted in statistically significantly prolonging time to second-line chemotherapy by 4.7 months when compared with placebo (19.7 months with Pazopanib versus 15 months with placebo, *p* = 0.0001) [23].

This was a well-designed study focusing on patient-centered endpoints and had a carefully designed PRO hypothesis. The context-specific PROs employed in the exploratory analyses support the benefits of prolongation of PFS to the patients.

The QoL measures in trials involving anti-angiogenics in the first-line treatment of ovarian cancer have been summarized in Table 1.

### 9.2. Cediranib in Platinum-Sensitive Ovarian Cancer

ICON6 (NCT00532194) was a phase III randomized trial of Cediranib in platinum-sensitive relapsed ovarian cancer [82]. There were three arms—chemotherapy and placebo (Arm A); Cediranib given with chemotherapy followed by placebo maintenance (Arm B); Cediranib given along with chemotherapy and continued as maintenance (Arm C). Published findings showed significant PFS extension with combination Cediranib and chemotherapy and as maintenance (2.3 months) [82] as well as OS improvement (7.4 months) [83]. Stark et al. [84] studied the HRQoL using EORTC QLQ-C30 and QLQ-OV28 during the first year of treatment as most patients experience good QoL at that point if they are not receiving chemotherapy and their cancer is controlled [84]. Using the QLQ-C30 tool, the primary study QoL endpoint was measured one year after study enrollment. Treatment groups consistent with survival and endpoints were compared [84]. Secondary outcomes included three hypotheses (early, mid and late) based on Cediranib’s mechanism of action, toxicity and on elements of maintenance therapy that can be important to patients [84]. These aimed to examine the effect of Cediranib at different time points in the course of treatment. The early hypothesis stated that ascites formation is related to angiogenesis; a rapid resolution is expected in patients on Cediranib, an anti-angiogenic agent [84]. The mid hypothesis stated that Cediranib provides quicker symptom relief owing to a rapid reduction in tumor bulk [84]. The late hypothesis stated that maintenance Cediranib may contribute to fatigue and result in impairment of performance of activities of daily living [84]. There was a 4.5 point increase in global QoL score in patients receiving maintenance Cediranib, which was not statistically and clinically significant; neither were there any significant differences in the secondary outcomes [84]. Cediranib neither improved signs and symptoms early on in the course of the disease (early hypothesis), nor at relapse (mid hypothesis) [84]. There was no significant difference in social functioning or fatigue while on maintenance Cediranib (late hypothesis) [84]. It was also shown that Cediranib could provide better QoL if diarrhea control was achieved [84]. This study indicated an incomplete association between patient report of diarrhea and clinical severity grading and also captured the distinct toxicity of voice change with Cediranib [84].

This was a well-designed study which carefully took into account the mechanism of action and toxicity of the drug, especially with long-term use. It also showed that not all adverse events affect QoL. Effective management of adverse events could provide better QoL.

### 9.3. Bevacizumab in Platinum-Resistant Ovarian Cancer

AURELIA (NCT00976911) was a phase III trial of chemotherapy (pegylated liposomal doxorubicin, topotecan or weekly paclitaxel) with or without bevacizumab [85]. Performance status, ascites, CA-125 levels, platinum-free interval and primary platinum resistance have been identified as prognostic factors in platinum-resistant ovarian cancer [85]. Adding bevacizumab to chemotherapy statistically significantly improved PFS by 3.3 months and objective response rate by 15.5% [85]. As symptom improvement is the major goal of platinum-resistant ovarian cancer, determining effects on HRQoL were the key secondary aim of AURELIA which was further described by Stockler et al. [86] EORTC QLQ-OV28 and FOSI questionnaires were used to conduct the QoL surveys. For the primary HRQoL endpoint, more patients who received Bevacizumab along with chemotherapy had a clinically and statistically significant improvement in the EORTC QLQ-OV28 abdominal/gastrointestinal symptom scale at week 8/9 (21.9% vs. 9.3%, *p* = 0.002) [86]. A 13.3% difference was observed while missing data were taken into consideration [86]. In the subgroup of symptomatic patients with a baseline score ≥15, there was a 16.9% difference, supporting the addition of Bevacizumab (29.6% vs. 12.7%) [86]. This was the first study to demonstrate that treatment can improve symptoms in platinum-resistant ovarian cancer; however, the benefits of Bevacizumab were not outweighed by its toxicity. Importantly, this study chose a more stringent 15% cut-off to reflect a meaningful clinical improvement.

Roncolato et al. analyzed whether QoL scores would be prognostic in platinum resistant ovarian cancer [87]. They focused on abdominal/gastrointestinal symptoms because ascites and peritoneal carcinomatosis significantly impact QoL. Patients were divided into four quartiles based on their EORTC QLQ-C30 scores. The first quartile was classified as good and the fourth as poor. They were then grouped into low- (quartile 1), medium- (quartile 2 and 3) and high- (quartile 4) risk categories [87]. It was shown that risk-stratifying patients based on physical function and abdominal symptoms correlated with their median OS [87]. Taken together, study findings showed that physical function and abdominal/gastrointestinal symptom scores improved predictions of OS in platinum-resistant recurrent ovarian cancer [87].

### 9.4. Pazopanib in Platinum-Resistant Ovarian Cancer

PACOVAR (NCT01238770) was a Phase I/II trial evaluating Pazopanib with metronomic Cyclophosphamide in platinum-resistant ovarian cancer [88]. The primary outcome was determination of the optimal dose of Pazopanib. It was shown that Pazopanib 600 mg taken orally daily, along with metronomic cyclophosphamide orally daily, is a feasible regimen in this previously treated patient population [88]. The secondary outcome measure included assessment of QoL overtime as defined by EORTC QLQ-C30 and OV28, which were given to the patients before treatment and after every three cycles as well as during follow-up. Unlike standard options, this trial involved oral therapy, a single blood test once a week and clinical assessment every four weeks. QoL was determined to be acceptable with no significant changes detected across domains (cognitive and emotional function, global health status) whilst on treatment; however, the study size is modest and, as such, cannot be generalized [88].

### 9.5. Sorafenib in Platinum-Resistant Ovarian Cancer

TRIAS was a randomized Phase II trial that showed that the combination of Sorafenib and Topotecan followed by Sorafenib maintenance in the setting of platinum-resistant disease resulted in prolongation of PFS and OS while compared to Topotecan and placebo [89]. PRO was a secondary endpoint in this study [89]. More closely examining global QoL scores (changes from baseline to end of chemotherapy) showed that results were globally stable for Sorafenib [89]. During chemotherapy, ability to perform life roles reduced clinically and statistically in both groups. Clinically significant worsening was noted only for dyspnea and diarrhea with Sorafenib [89]. Study limitations were notable as questionnaire completion was not compulsory at visits, which resulted in modest data collected and longitudinal analysis was not possible [89].

## 10. QoL Data from PARP Inhibitor Trials in Ovarian Cancer

This section summarizes the QoL data from landmark trials involving PARP inhibitors in ovarian cancer, in the front-line and recurrent settings.

### 10.1. Olaparib Maintenance Therapy in Platinum-Sensitive Relapsed Ovarian Cancer: Study 19 and SOLO2/ENGOT Ov-21

Study 19 (NCT00753545) was a seminal Phase II study of Olaparib maintenance in women with platinum-sensitive recurrent ovarian cancer [90]. It explored topics around HRQoL [69] and findings showed no significant between-group differences in disease-related symptoms or improvement in HRQoL [69], which were measured using the FACT-O questionnaire, FOSI and TOI [69]. The time of the worsening of each of these endpoints was shorter with Olaparib than with placebo; however, the difference was not statistically or clinically significant [69]. This study showed that Olaparib had no adverse impact on HRQoL [69]; complemented by high compliance and low adverse event-related discontinuation rates [69]. The limitations of this study include no collection of HRQoL data beyond disease progression and data about delays in time to first subsequent therapy or death (TFST).

SOLO2/ENGOT-Ov21 (NCT01874353) was a double-blind randomized Phase III trial, which showed Olaparib tablet maintenance resulted in significant prolongation of PFS in platinum-sensitive recurrent ovarian cancer [91]. The most common adverse events experienced by patients were fatigue, nausea and vomiting, all of which were low grade and improved with time [91]. Friedlander et al. analyzed the QoL outcomes of this study [70]. The authors hypothesized that Olaparib would not have a decremental effect on HRQoL and that it would be supported by a meaningful increase in patient-centered endpoints [70]. This was confirmed by the final results, which in fact showed that HRQoL remained stable while on olaparib maintenance and significant increases in the duration of good quality of life (TWiST) and QAPFS were reported [70].

This study showed that Olaparib maintenance did not have a decremental effect on HRQoL [70]. In addition, also seen were clinically significant improvements in patient-centered endpoints such as TWiST and QAPFS [70]. In the final analysis, maintenance Olaparib provided an unprecedented improvement of 12.9 months in median overall survival vs. placebo [92]. Updated HRQoL data are awaited.

### 10.2. Maintenance Olaparib in Patients with Newly Diagnosed Advanced Ovarian Cancer

SOLO1 (NCT01844986) was a Phase III trial of Olaparib as front-line maintenance in patients with newly diagnosed advanced ovarian cancer with *BRCA* mutations who had a complete or partial clinical response to chemotherapy [93]. The primary endpoint was PFS. The use of maintenance Olaparib provided a substantial PFS benefit, lowering the risk of disease progression or death by 70% [93]. Use of maintenance Olaparib did not have a detrimental effect on QoL [93]. Furthermore, Wolford et al. have compared the cost-effectiveness of Olaparib in upfront (SOLO1) versus the recurrent maintenance setting (SOLO2) and shown that upfront maintenance therapy is more cost-effective [94].

### 10.3. Niraparib in Patients with Platinum-Sensitive Recurrent Ovarian Cancer

NOVA (NCT018472274) was a randomized, double-blind, phase III trial of Niraparib maintenance in platinum sensitive disease. Patients were grouped based on the presence or absence of a germline *BRCA* mutation and the type of non-gBRCA mutation [4]. Niraparib maintenance conferred a significantly longer PFS regardless of the *BRCA* or homologous recombination deficiency (HRD) status [4].

Do the adverse effects of Olaparib affect QoL? Does the effect on QoL balance the prolongation of PFS? Oza et al. have conducted an in-depth study which answers these questions [66]. Using the FOSI and EQ-5D-5L questionnaires, PRO assessment was conducted at the time of trial entry and at multiple time points while on study [66]. For patients who discontinued treatment, PROs were collected at the time of discontinuation and 8 weeks post-progression [66]. Findings show that the baseline mean FOSI score was similar between the two groups, and overall QoL scores remained stable while on treatment with niraparib [66]. The most common adverse events during the screening period were fatigue (79%, with 18% reporting severe lack of energy), pain (44%), and nausea (22%) [66]. All symptoms, except nausea, either remained stable or improved over time in patients on Niraparib [66]. The most common grade 3 or 4 toxicities in the niraparib group were hematological, characterized by thrombocytopenia (34%), anemia (25%) and neutropenia (20%) [66]. Disutility analyses showed there was no significant QoL impairment associated with these toxic effects [66]. Therefore, the NOVA trial showed that Niraparib has no significant negative effect on QoL in the setting of recurrent ovarian cancer, and provides additional evidence supporting the addition of Niraparib as a standard of care [66].

### 10.4. Rucaparib in Patients with Platinum-Sensitive Recurrent Ovarian Cancer

ARIEL3 (NCT01968213) was a Phase III randomized trial of Rucaparib maintenance in platinum-sensitive recurrent ovarian cancer. Rucaparib significantly improved PFS across all primary analysis groups [95] as well as time to first subsequent therapy (TFST), time to PFS on subsequent line of treatment or death (PFS2) and time to second subsequent therapy (TSST) versus placebo [96]. Patient-centered outcomes of QAPFS and QTWiST were calculated using the EQ-5D-3L questionnaire. Regardless of the loss of heterozygosity (LOH ) status or the presence or absence of a BRCA mutation, Rucaparib significantly prolonged QAPFS and QTWiST [97].

### 10.5. Maintenance Veliparib in Patients with Newly Diagnosed Advanced Ovarian Cancer

VELIA/GOG-3005 (NCT02470585) was a Phase III trial examining the addition of Veliparib to first-line chemotherapy, and subsequently continued as single agent maintenance treatment in women with newly diagnosed advanced high-grade serous ovarian carcinoma [98]. Across all trial populations, a regimen of chemotherapy and Veliparib followed by Veliparib maintenance therapy led to significantly PFS than chemotherapy alone [98]. In all populations, the mean change from baseline in the NFOSI-18 Disease-Related Symptom scores indicated improvement over time, particularly after completion of chemotherapy [98].

The QoL measures in trials involving PARP inhibitors as maintenance in platinum-sensitive recurrent ovarian cancer have been summarized in Table 2.

### 10.6. Niraparib in Patients with Newly Diagnosed Advanced Ovarian Cancer

PRIMA/ENGOT-OV26/GOG-3012 (NCT02655016) was a Phase III trial of Niraparib in patients with advanced high grade serous or endometrioid ovarian cancer, who have partial or complete response to front-line chemotherapy [99]. This study showed Niraparib conferred a PFS benefit of 5.6 months, irrespective of HRD. The trial employed FOSI, EQ-5D-5L and EORTC-QLQ-C30/OV28 instruments, which did not indicate a significant difference in HRQoL scores [99].

### 10.7. Missed Opportunities, Lessons Learned and Initiatives Promoting High-Quality PRO Research

Friedlander et al. [100] have elaborated on the missed opportunities and lessons learnt in PROMs in ovarian cancer clinical trials, citing several illustrations [80,101,102,103,104,105] summarized in Table 3. Lost opportunities include (a) no clear pre-specified PRO hypotheses; (b) PRO endpoints not included/limited analysis of HRQoL; (c) insensitive PRO endpoint selection; (d) collection of poor-quality PRO data not suitable for analysis; (e) differences in PROs ignored; (f) poor reporting quality. Together, these initiatives promote high-quality PRO research.

### 10.8. Measuring QoL in Ovarian Cancer Patients Receiving Immunotherapy

Immune-related adverse events are unique to immunotherapeutic agents. Immunotherapeutic agents include inhibitors of programmed cell death protein 1 (PD-1), its ligand, programmed death-ligand 1 (PD-L1) and cytotoxic T-lymphocyte antigen-4 (CTLA-4) [106]. The class effects of this group of drugs are collectively labelled as immune-related adverse events [106]. They are mediated by auto-reactive T cells and tissue-infiltrating immune cells which can have a significant impact on HRQoL.

Hall et al. [107] performed a systematic review to examine PROs and HRQoL among cancer patients receiving immune checkpoint modulators (ICMs) as compared to other anti-cancer therapies. This included patients with melanoma, lung, genitourinary and head and neck cancers. Of the seven PROMs, EORTC QLQ-C30 was the most commonly used (80%). While ICMs are associated with significant adverse effects potentially involving multiple organ systems, patient-reported HRQoL did not differ greatly when compared to other therapies [107]. As ICMs share few common toxicities with chemotherapy, established PROMs are not equipped to capture the true impact on HRQoL; thus, highlighting a need for novel PROM development within the ICMs space [107]. Hansen et al. [106] have developed the Functional Assessment of Cancer Therapy—Immune Checkpoint Modulator (FACT-ICM), which is a 25-item toxicity subscale to measure QoL in cancer patients who are treated with ICMs. Four out of the 37 patients interviewed in the development of this questionnaire had gynecological cancers. Adverse events, such as cutaneous toxicity, cough, taste disturbance and fever and chills, which are not captured in EORTC QLQ-C30, have been included in this tool. This PRO tool has been designed keeping in line with the US-FDA guidance which emphasizes meaningful patient input right from the onset of treatment [10]. Once this subscale has completed reliability and validity testing, it can be integrated into ovarian cancer trials of ICMs as well.

## 11. Discussion

Patients volunteering to participate in clinical trials and inclusion of PROs is an important acknowledgement of the importance of patient-centered research, which should reflect their perspective of treatment benefit and experience of side-effects. It is important to keep in mind that the balance of toxicity and benefit for each patient is unique and personal. There is room for improvement in QoL data collection and presentation in trials, with early planning of the inclusion of PRO data guided by a priori hypothesis. It is sometimes challenging to compare trials of the same class of compound, or even the same agent, which may use different instruments or analytic techniques. Some standardization of implementation and analysis by the academic community may help in this effort. It is also critically important to add the context and relevance of other endpoints such as OS, PFS and clinical benefit. Intriguingly, QL may also be an important predictive factor in its own right [108]. Use of open-ended questions regarding anxiety, depression and other mental health issues and patients’ perspectives of patient safety will help make PRO data informative. Checklists have been provided by CONSORT-PRO [109] and SPIRIT-PRO [110] to facilitate optimal reporting of randomized controlled trials where PROs are primary or secondary endpoints. As we usher in the next generation of clinical trials assessing combinations of immunotherapeutic agents, immunotherapy in combination with chemotherapy—with or without anti-angiogenics and PARP inhibitors—improving QoL collection and presentation in trials, will heighten the relevance of the study results to patients, improve content and construct validity of measures [111,112,113]. As we increase complexity, some of the possible negative consequences, including added financial burden and the need to enhance staff resources to establish and maintain engagement, need to be factored in. In addition, answering questionnaires might make patients feel that there is a “work burden” in participating in a trial and this could be a dis-incentive.

Applications of knowledge of PROMs in important, clinically relevant scenarios include analysis of PRO/HRQoL in management of malignant bowel obstruction and ascites, which is useful in framing strategies to reduce visits to the emergency department. Another potential area of real-world analysis may be the comparison of HRQoL/PRO in groups of patients on different PARP inhibitors (their perception about the choice of PARPi and its toxicity profile).

## 12. Conclusions

PROs are important measures of benefits and risks of ovarian cancer treatment. They add value to survival data such as OS and PFS, enhance patient centeredness, physician–patient communication and help guide in decision making. As we continue to add to our treatment armamentarium, dynamic and continuous development of PROMs based on context and the type of treatment will be essential. Though beyond the scope, attention to capturing PROs, HRQoL and use of digital and telemedicine should be considered. In addition, addressing those with poor literacy to capture QoL of our most vulnerable is warranted.

## Figures and Tables

**Table 1 cancers-12-03296-t001:** Compare and contrast quality of life (QoL) measures in trials involving anti-angiogenics in the first-line treatment of ovarian cancer [23,46,47,77,78,79,80].

Trial	QoL Measures Used
GOG 218 [46,47,77]	Relationship between baseline and serial QoL measurements and PFS and OS
Chemotherapy alone versus	
Chemotherapy + Bevacizumab versus Chemotherapy + Bevacizumab followed by Bevacizumab maintenance	
ICON7 [78,79,80]	Comparing global QoL at baseline and time of completion of chemo and at week 54 on treatment
Chemotherapy alone versus	
Chemotherapy + Bevacizumab followed by Bevacizumab maintenance	
AGO-OVAR 16 [23,81] Maintenance Pazopanib versus placebo	Mean change in scores in the “on treatment” patients (patients on Pazopanib) during maintenance phase
	Impact of adverse events on HRQoL
	QAPFS, impact of specific symptoms and progressive disease on HRQoL and time to second-line chemotherapy

Abbreviations: QoL, quality of life; PFS, progression-free survival; OS, overall survival; HRQoL, health-related quality of life; QAPFS, quality-adjusted progression-free survival.

**Table 2 cancers-12-03296-t002:** Comparisons and contrasts of QoL measures in trials involving PARP inhibitors as maintenance in platinum-sensitive recurrent ovarian cancer [4,66,69,70,91].

Trial	QoL Measures Used
Study 19 [69]	Differences between groups receiving Olaparib and placebo with regard to disease-related symptoms or rates of improvement in HRQoL
SOLO2 (Olaparib) [70,91]	Change from baseline in TOI score during the first twelve months of the study
	TWiST
	QAPFS
NOVA (Niraparib) [4,66]	Assess the effect of hematological toxicity on QoL
ARIEL3 (Rucaparib) [95,96,97]	QTWiST
	QAPFS
VELIA (Veliparib) [98]	NFOSI-18 Disease-Related Symptom

Abbreviations: QoL, quality of life; HRQoL, health-related quality of life; TOI, trial outcome index; TWiST, time without symptoms or toxicity; QAPFS, quality-adjusted progression-free survival; NFOSI, National Comprehensive Cancer Network Functional Assessment of Cancer Therapy Ovarian Symptom Index-18.

**Table 3 cancers-12-03296-t003:** Missed opportunities in ovarian cancer trials.

Trial Details	Results	Available PRO Data	Other Details	Missed Opportunity and Lessons Learned
Colombo N et al. [101] Patupilone (*P*) versus Pegylated liposomal doxorubicin (PLD) Phase III	OS (primary endpoint): 13.2 months (*P*) vs. 12.7 months (PLD) PFS: 3.7 months in both arms Overall response rates: 15.5% (*P*) vs. 7.9% (PLD) Negative trial	QoL was briefly mentioned in online appendix only No difference in mean scores of global HRQoL	Frequently observed toxicities: Diarrhea (85%) and peripheral neuropathy (39%) in (*P*) Stomatitis/mucositis (43%) and hand-foot syndrome (41.8%) in (PLD)	No PRO endpoints/limited analyses of HRQoL Measurement of symptom benefit can strengthen the PFS benefit in a negative trial Suggested PROMs: (1) Duration of symptom control (2) Proportion of patients symptomatic at baseline (3) Time until definitive quality of life score deterioration (TUDD)
OCEANS [103] Chemotherapy + Bevacizumab (BV) versus Chemotherapy + Placebo (PL) Phase III	PFS: 12.4 months (BV) vs. 8.4 (PL) months Duration of response: 10.4 months vs. 7.4 months	HRQoL was not measured No PRO endpoints	Grade 3 or higher hypertension (17.4% vs. <1%) and proteinuria (8.5% vs. <1%) were more frequent in BV arm	No PRO endpoints/limited analyses of HRQoL PRO endpoints would have strengthened the PFS benefit Can possibly lead to a labelling indication Suggested PROMs: (1) Time course of toxicity (2) Time to deterioration of HRQoL (3) HRQoL on treatment and at progression
ICON 7: [80,104] Paclitaxel + Carboplatin (PC) vs. Paclitaxel + Carboplatin + Bevacizumab followed by maintenance Bevacizumab (PCB) Phase III	PFS at 36 months: 20.3 months (PCB) vs. 20.3 months (PC)	No special attention to PRO endpoints in high-risk group	Grade 2 or higher hypertension was more frequent in Bev (18% vs. 2%)	Lack of clinically important/sensitive PRO endpoints: Context-specific PRO endpoints to interpret efficacy from patient perspective Suggested context-specific PROMs: QTWiST, QAPFS, compliance with treatment, patient trade-offs (multiple IV infusions, toxicities)
Reed NS et al. [105] Carboplatin versus Treosulfan in older women unfit to receive single agent carboplatin Phase III	Overall response rates: Carboplatin (49%) vs. Treosulfan (29%) Median time to progression: 10 months (Carboplatin) vs. 5 months (Treosulfan)	QoL was a secondary endpoint and strictly requested in the protocol Limited QoL data (small proportion of patients filled in questionnaires, exact number not mentioned in results)	Thrombocytopenia due to Treosulfan: Frequent dose reductions and delays	Poor data quality: Can be due to lack of compliance by the participants or lack of motivation by the physicians Suggested measures: Quality assurance/quality control in HRQoL data collection
Kaye SB et al. [102] Olaparib (O) 200 mg twice daily versus Olaparib (O) 400 mg twice daily versus Pegylated liposomal doxorubicin (PLD) in BRCA-related recurrent ovarian cancer Included platinum-sensitive and platinum-resistant cohorts Phase II	No significant difference in PFS: 8.8 months (O) vs. 7.1 months (PLD) Negative trial	No significant differences in improvement/worsening rates between groups (FOSI and TOI scores) Higher improvement was noted for Olaparib 400 mg twice daily vs. PLD for total FACT-O score (odds ratio, 7.23; 95% CI, 1.09 to 143.3; *p* = 0.039).		Ignoring differences in PRO endpoints while determining clinical trial relevance: Suggested remediations: Pre-planning PRO hypothesis, examining patient preferences (capsules over intravenous infusions), patient-reported toxicity, trajectory of toxicities rather than cumulative toxicities This would have led to faster development of Olaparib program

Abbreviations: PRO, patient-reported outcomes; OS, overall survival; HRQoL, health-related quality of life; PLD, pegylated liposomal doxorubicin; PROMs, patient-reported outcome measures; QTwiST, quality-adjusted time without symptoms or toxicity; QAPFS, quality-adjusted progression-free survival; TOI, trial outcome index; FOSI, FACT-O symptom index.

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
