# Peer review of "Measuring Quality of Life in Ovarian Cancer Clinical Trials—Can We Improve Objectivity and Cross Trial Comparisons?"

_cancers, 2020, doi:10.3390/cancers12113296_

Round 1
Reviewer 1 Report
This is a nicely organized and comprehensive review. Outcomes from multiple studies have been described. The review highlights the need for more complete development of hypotheses and inclusion of appropriate tools to ensure PRO are fully considered.
Author Response
We would like to extend our gratitude to Reviewer 1 for their thoughtful and thorough review of our manuscript. No additional changes were requested; however, for minor English language and style are fine/minor spell check required we have requested editorial assistance to ensure accuracy.
Reviewer 2 Report
Bhat et al., reported focused on quality of life of patient who is participating ovarian cancer clinical trials. This is well written review paper including comprehensive clinical trials. Overall merit for readers are high.
Author Response
We would like to extend our gratitude to Reviewer 2 for their thoughtful and thorough review of our manuscript. No additional changes were requested; however, for minor English language and style are fine/minor spell check required we have requested editorial assistance to ensure accuracy.
Reviewer 3 Report
I believe this offers an incredibly important commentary on the importance of trial development and patient centered treatments. The authors do an excellent job of analyzing and presenting the background of these measures and integrating the results of QOL and hrQOL measures used in key landmark practice changing trials. Key measures were included and explained well. The organization is clear and flows well. Even those unfamiliar with the various measure will leave with a better understanding. This review will assist researchers and trialists to better understand the process of choosing measures and more importantly specific endpoints for the measures, not just collecting and analysis post. In addition, for clinicians treating ovarian cancer, this work also presents a wonderful review that can be used to frame and review treatment choices with patients. I would suggest a distillation of this to the patient community would be very welcome. Though beyond the scope, attention to capturing PROs, hrQOL and use of digital and telemedicine should be considered. In addition, addressing those with poor literacy to capture QOL of our most vulnerable would be interesting. Thank you for your work
Author Response
We would like to extend our gratitude to Reviewer 3 for their thoughtful and thorough review of our manuscript. We have included reference in the revised manuscript to now include the following information in the conclusion:
"Though beyond the scope, attention to capturing PROs, hrQOL and use of digital and telemedicine should be considered. In addition, addressing those with poor literacy to capture QOL of our most vulnerable is warranted."
Furthermore, for minor 'English language and style are fine/minor spell check required' we have requested editorial assistance to ensure accuracy.